

**Molecular Composition and Photochemical Evolution of Water**
**Soluble Organic Carbon (WSOC) Extracted from Field Biomass**
**Burning Aerosols using High Resolution Mass Spectrometry**
Jing Cai[1,2], Xiangying Zeng[1], Guorui Zhi[3], Sasho Gligorovski[1], Guoying Sheng[1],
Zhiqiang Yu[1,*], Xinming Wang[1], Ping'an Peng[1]
[1]*State Key Laboratory of Organic Geochemistry, Guangdong Key Laboratory of*
*Environment and Resources, Guangzhou Institute of Geochemistry, Chinese*
*Academy of Sciences, Guangzhou, 510640, China*
[2]*University of Chinese Academy of Sciences, Beijing, 100049, China*
[3]*State Key Laboratory of Environmental Criteria and Risk Assessment, Chinese*
*Research Academy of Environmental Sciences, Beijing, 100012, China*
*corresponding author: Dr. Zhiqiang Yu
Tel: +86-13728068752
Fax: +86-20-85290288
E-mail: zhiqiang@gig.ac.cn



**ABSTRACT**
Photochemistry plays an important role in the evolution of atmospheric water
soluble organic carbon (WSOC), which dissolves into clouds, fogs and aerosol liquid
water. In this study, we examined the molecular composition and evolution of a
WSOC mixture extracted from fresh biomass burning aerosols upon photolysis,
using direct infusion electrospray ionization high-resolution mass spectrometry
(ESI-HRMS) and liquid chromatography coupled with mass spectrometry
(LC/ESI-HRMS). For comparison, two typical phenolic compounds (i.e., phenol and
guaiacol) emitted from lignin pyrolysis in combination with hydrogen peroxide
($H_2O_2$) as a typical OH radical precursor, were exposed to simulated sunlight
irradiation. The photochemistry of both, the phenols (photo-oxidation) and WSOC
mixture (direct photolysis) can produce a series of highly oxygenated compounds
which in turn increases the degree of oxidation of organic composition and acidity of
the bulk solution. In particular, the LC/ESI-HRMS technique revealed significant
photochemical evolution on the WSOC composition, e.g., the photodegradation of
low oxygenated species and the formation of highly oxygenated products. We also
tentatively compared the mass spectra of photolytic time-profile extract with each
other for a more comprehensive description of the photolytic evolution. The
calculated average oxygen-to-carbon (O/C) ratios of oxygenated compounds in bulk
extract increases from $0.38\pm0.02$ to $0.44\pm0.02$ (mean$\pm$standard deviation) while
the intensity (S/N)-weighted average O/C ($O/C_w$) increases from $0.45\pm0.03$ to $0.53$
$\pm0.06$ as the time of irradiation extends from 0 to 12h. These findings indicate that



the water soluble organic fraction of fresh combustion-derived aerosols have the
potential to form more oxidized organic matter, accounting for the highly
oxygenated nature of atmospheric organic aerosols.
**1 INTRODUCTION**
Water-soluble organic carbon (WSOC) comprises a significant fraction of
atmospheric aerosols, accounting for 20–80% of total organic carbon (OC) (Krivacsy
et al., 2001; Wozniak et al., 2008; Fu et al., 2015; Xie et al., 2016).WSOC is directly
involved in the formation of cloud condensation nuclei (CCN) by modifying the
aqueous chemistry and surface tension of cloud droplets (Graham et al., 2002;
Nguyen et al., 2012; Zhao et al., 2013; McNeill 2015). Despite its significance, little
is known about the chemical composition and sources of WSOC, with less than
10–20% of the organic mass being structurally identified (Cappiello et al., 2003; Fu
et al., 2015). Biomass burning is a well-known emission source of WSOC (Anastasio
et al., 1997; Fine et al., 2001; Graham et al., 2002; Mayol-Bracero et al., 2002;
Gilardoni et al., 2016).Although the composition varies with fuel type and
combustion conditions (Simoneit 2002; Smith et al., 2009), the WSOC mixture often
covers a common range of polar and oxygenated aromatic compounds (Graham et al.,
2002; Mayol-Bracero et al., 2002; Duarte et al., 2007; Chang and Thompson 2010;
Yee et al., 2013; Gilardoni et al., 2016) with molecules incorporating different
numbers of functional groups like COOH, C=O, CHO, COH, COC, $CONO_2$, CNH,
and/or $CONH_2$ groups (Graham et al., 2002). In particular, lignin pyrolysis often
yields a large amount of aromatic alcohols, carbonyl, and acid compounds



(Mayol-Bracero et al., 2002; Chang and Thompson 2010; Gilardoni et al., 2016).
Once dissolved into cloud, fog, and even aerosol liquid water, these substances can
undergo aqueous-phase reactions to affect aerosol evolution processes under sunlight
irradiation, and produce low-volatility species, which have the potential to form
secondary organic aerosol (SOA) after water evaporation (Graham et al., 2002;
Cappiello et al., 2003; Duarte et al., 2007; Sun et al., 2010; Yu et al., 2014).
Field and laboratory studies have demonstrated that aqueous photochemical
processes contribute significantly to the aqueous SOA formation from biomass
burning precursors and the evolution of smoke particles (Sun et al., 2010; Lee et al.,
2011; Kitanovski et al., 2014; Yu et al., 2014; McNeill 2015; Gilardoni et al., 2016).
Gilardoni et al. (2016) observed aqueous SOA formation in both fog water and wet
aerosols, resulting in an enhancement in the oxidized OA, and following
atmospheric aging the overall O/C ratios of aerosols has also increased. In laboratory
studies, phenols and methoxyphenols (important biomass burning emerged
intermediates) are often used as SOA precursors to examine the photochemical
evolution in aqueous environment and aerosol-forming potential under relevant
atmospheric conditions (Chang and Thompson 2010; Sun et al., 2010; Smith et al.,
2014; Yu et al., 2014, Vione et al., 2019). The corresponding photochemical products
formed through hydroxylation, oligomerization, and fragmentation typically cover a
series of low-volatility and highly oxygenated species. For instance, the
methoxyphenol-derived SOA are proposed as a proxy for atmospheric humic-like
substances (HULIS) (Ofner et al., 2011; Yee et al., 2013). Other compounds emitted



from lignin pyrolysis, e.g., aromatic alcohol, carbonyl, and carboxylic species
retaining the phenyl ring have also been found to produce colored products via
aqueous photo-oxidation, which may become a part of HULIS (Chang and
Thompson 2010, Huang et al., 2018). In addition, photochemical processing of
common water-soluble aliphatic compounds such as aldehydes (Lim and Turpin
2015), polyols (Daumit et al., 2014), and organic acids (Griffith et al., 2013) in
aqueous solution can also lead to the formation of oligomers, highly oxygenated and
multifunctional organic matter (McNeill 2015).
In recent years, high resolution mass spectrometry (HRMS) has been commonly
applied to study the organic molecular composition in cloudwater (Zhao et al., 2013;
Boone et al., 2015), fogwater (Cappiello et al., 2003), rainwater (Altieri et al., 2009a;
Altieri et al., 2009b), laboratory-generated SOA (Bateman et al., 2011; Romonosky
et al., 2015; Lavi et al., 2017), and field-collected aerosol samples (Laskin et al.,
2009; Lin et al., 2012a; Lin et al., 2012b; Kourtchev et al., 2013). It has also been
used in time-profile observations of the photochemical evolution of aqueous extracts
from laboratory-generated SOAs (Bateman et al., 2011; Romonosky et al., 2015).
However, direct infusion MS methods are prone to ion suppression caused by other
organic species, inorganic salts, and adduct formation (Kourtchev et al., 2013).
Therefore, HRMS coupled with LC might be another complementary powerful tool
for relieving ion suppression (Kourtchev et al., 2013; Wang et al., 2016). It could
also provide more information enabling the identification of possible isomers from
the ions with same mass-to-charge ratio (m/z).



To our knowledge, the aqueous photochemical evolution of WSOC extracted from
real ambient aerosols has not been studied in detail at the molecular level. The
present study is focused on a further analysis of the previously studied field collected
samples by Cai et al. (2018). Here, the main goal is to investigate the molecular
characteristics of water-soluble organic molecules by the photochemical evolution
using ESI-HRMS and LC/ESI-HRMS performed in negative ionization mode. The
photochemistry of phenol and guaiacol was evaluated under laboratory conditions as
well, and used as a reference.
**2 EXPERIMENTAL SECTION**
**2.1 Particulate sample collection and preparation of aqueous extracts**
Fresh straw-burning aerosols were collected during the summer harvest season of
2013, at rural fields in north China (Cai et al., 2018). Briefly, the selected samples
used for HRMS analysis were collected from two sampling sites, located at rural
fields in Wenxian in Henan Province (noted: HNWX) and Daming county in Hebei
Province (HBDM). The selected sampling sites were mainly affected by heavy smog
from straw burning (Figure 1). As described in Cai et al. (2018), particulate matter
($\leq 2.5\mu m$) was sampled by a portable particulate sampler (MiniVol TAS, AirMetrics,
USA), with quartz filters (47mmin diameter, QMA, Whatman, UK) baked at 600°C
for 6 hours before sampling.
The preparation of straw-burning particle extracts and measurements for carbon
content including organic carbon (OC), elemental carbon (EC) and WSOC were





described in detail in Cai et al. (2018). A pH meter (Mettler Toledo SevenEasy$^{TM}$
S20) calibrated at pH 4.00 and 6.86 was applied for extract pH measurements. Prior
to analysis the extract solutions were stored at -20°C in the dark. To reduce the
WSOC mass loss, the desalting treatment (e.g., solid phase extraction (SPE)) was not
performed on these samples.

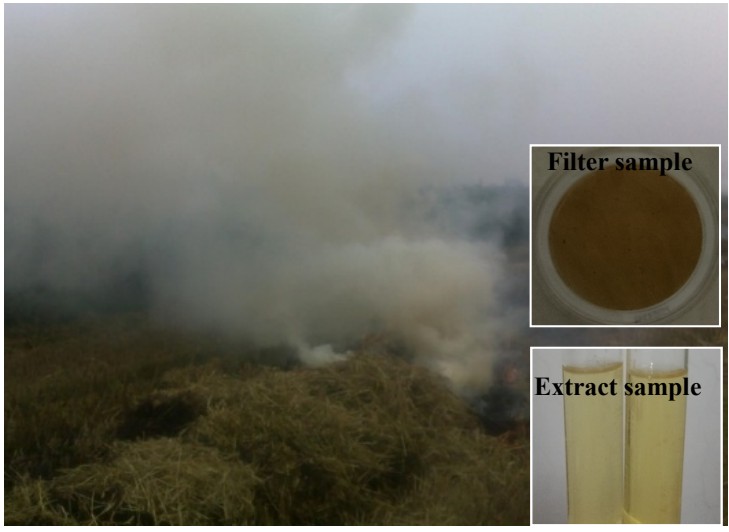

**Figure 1. One field site at Daming, Hebei province, China, for sampling the aerosols affected by biomass burning.**

**2.2 Laboratory observation of the direct photolysis of WSOC extracts**
A 12-hour direct photolysis of particle extracts was performed in a photoreactor
(BL-GHX-V, Bilon Instruments Co. Ltd., China, see Figure S1) (Cai et al., 2018). In
the wavelength range of 310-400 nm relevant to the boundary layer of the
atmosphere, the actinic flux of the lamp is about 5 times stronger than the solar
actinic flux, meaning that the spectral evolution via the 12-hour simulated solar
irradiation might be equal with the effect caused by actual sunlight irradiation with a



duration of at least 60 hours. As described in Cai et al. (2018), the water extraction
resulted in a dilution of the collected organic compounds, however, the ratio of the
water mass to PM2.5 mass for extraction was compatible with the ratio of water mass
to WSOC content in cloud water, indicating that the present aqueous extracts are
relevant to the atmospheric cloud water condition (Li et al., 2017).
In the experimental section of phenol photochemistry, initial solution of 0.1 mM
phenol and 0.1 mM guaiacol in combination with an OH radical precursor (0.1 mM
$H_2O_2$), were prepared in ultra-pure water (Milli-Q, Milipore). The pH of the solution
was adjusted to 5 with 0.1 M sulfuric acid ($H_2SO_4$), which is usually relevant to the
acidity in fog and cloud waters (Collet et al., 1998, Fahey et al., 2005). The prepared
solution and reference blank were irradiated by simulated sunlight irradiation with a
duration of 4 hours. Hereby, we mainly focus on acquiring the chemical
characteristics of aqueous products of phenols, and tentatively identify some
biomarkers (e.g., phenolic dimers) whether they exist in the real biomass burning
particulate samples.
**2.3 Sample analysis**
The direct infusion MS analysis was conducted using a Thermo Scientific
Orbitrap Fusion Tribrid mass spectrometer equipped with a quadrupole, orbitrap, and
linear ion trap mass analyzers, with a heated ESI source. To assist in ionization and
desolvation, the aqueous extract was diluted to a 1:1 mixture of acetonitrile and
sample by volume. The full scan mass spectra were acquired in negative ionization
mode, with a resolution of 120 000 at *m/z* 200 for the orbitrap analyzer and a mass



scan range of m/z 50-750. Before determination, the orbitrap analyzer was externally
calibrated for mass accuracy using Thermo Scientific Pierce LTQ Velos ESI
calibration solution. The direct infusion parameters were as follows: sample flow
rate 5μl min$^{-1}$; capillary temperature 300°C; S-lens RF 65%; spray voltage -3.5kV;
sheath gas, auxiliary gas, and sweep gas flows were 10, 3, and 0 arbitrary units,
respectively. Data collecting was performed when the intensity of the total ion
current (TIC) maintained constant with an RSD＜5%. At least 100 data points (mass
spectral scans) were collected for each test sample, and the each exported mass
spectra for analysis was derived from the average result of 100 spectrums.
The LC/ESI-HRMS analysis operated in negative ionization mode was performed
using a U3000 system coupled with a T3 Atlantis C18 column (3μm; 2.1×150mm;
Waters, Milford, USA) and an Orbitrap Fusion MS. A 10 μL extract was injected,
with a flow rate of 0.2 ml min$^{-1}$ for the mobile phase, which consisted of $H_2O$ (A)
and acetonitrile (B). The gradient applied was 0-5 min 3% B; 5-20 min from 3 to 95%
(linear), and kept for 25 min at 95%; and 45-50 min from 95 to 3%, and held for 10
min at 3% (total run time 60 min).
**2.4 Data processing**
Mass spectral peaks with three times larger than the signal to noise ratio (S/N)
were extracted from the raw files. Peaks in both sample and blank spectra were
retained if their intensity in the former was five times larger than in the latter. A
molecular assignment based on the accurate mass was performed using Xcalibur
software (V3.0 Thermo Scientific) with the following constraints: $^{12}C{\leq}50$, $^{13}C{\leq}1$,





$^1H\leq100$, $^{16}O\leq50$, $^{14}N\leq4$, $^{32}S\leq1$, and $^{34}S\leq1$. All mathematically possible elemental
formulas, with a mass tolerance of ±3ppm were calculated. Elemental formulas
containing $^{13}C$ or $^{34}S$ were checked for the presence of $^{12}C$ or $^{32}S$ counterparts,
respectively. If they were not matched with the corresponding monoisotopic
formulas, then the assignment with next larger mass error was considered. Isotopic
and unassigned peaks were excluded from further analysis.
Ions were also characterized by the number of rings plus double bonds (i.e.,
double bond equivalents (DBE)), which were calculated as: $DBE=c-h/2+n/2+1$ for
an elemental composition of $C_cH_hO_oN_nS_s$. The assigned formula was additionally
checked with the nitro-rule. For ambient samples, based on the presence of various
elements in a molecule, the identified elemental formulas were classified into several
main compound classes: CHO (i.e., molecules containing only C, H, and O atoms),
CHOS, CHON, and CHONS, and others including CHN and CHS. In the present
study, because the detected water-soluble ions almost were below m/z 400, we
focused our molecular analysis on m/z 50-400.
**3 RESULTS AND DISCUSSION**
**3.1 Mass spectral characteristics of WSOC in biomass burning particulate**
The PM2.5 concentration in present straw burning smoke samples ranges from
6.46 to 28.03 mg m$^{-3}$ (Table S1). OC is the major component of the collected PM2.5
with a proportion of 50.9 ± 7.6% (mean ± standard deviation), whereas EC
represents a negligible fraction (average 1.3±0.4%). Meanwhile, WSOC accounts
for 35.5±7.5% of OC in the tested samples.



Four extract samples (HNWX-1, HNWX-2, HBDM-1 and HBDM-2) (Table S1)
analyzed using high resolution mass spectrometry showed similar patterns in mass
distribution of water-soluble molecular species that mainly range from 50 to 400 Da.
A group of reconstructed mass spectrum (abstracted blank) for two representative
samples HNWX-1, and HBDM-1 is shown, as an example, in Figure 2. In mass
range 50-400 Da, there were $827\pm44$ molecular formulas identified throughout the
all samples, and most of the formulas (above 75%) were overlapped between these
analyzed samples. The classification features of assigned compounds for analyzed
extracts are shown in Table S2. In the assigned formulas, CHO compounds were the
most abundant group, accounting for $59.2\pm2.2\%$ of the total assignments, followed
by CHON ($35.0\pm2.2\%$). These results are consistent with previous observations of
laboratory-generated biomass burning aerosol (Smith et al., 2009) and field
particulate samples influenced by biomass combustion (Kourtchev et al., 2016),
although the differences of biomass varieties, extracted solvent, and HRMS
techniques between present and previous studies. On the other hand, CHOS and
CHONS compounds contributed with less than 5% to the total assignment. A number
of studies have shown the wide presence of organosulfates and
nitrooxy-organosulfates in urban (Lin et al., 2012b; Wang et al., 2016), rural (Lin et
al., 2012a), and forest aerosols (Kourtchev et al., 2013), and even in cloudwater
(Boone et al., 2015); however, most of these compounds were not observed in our
negative mass spectra. This could be accounted for by the low extent of aerosol
evolution, due to the limited oxidation conditions available for the formation of

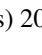



organosulfates and nitrooxy-organosulfates in fresh smoke aerosols. For example,
laboratory studies have observed the significant formation of organosulfates via
photooxidation in the presence of acidic sulfate aerosol (with significant level of $SO_2$
concentration) (Surratt et al., 2007; Surratt et al., 2008). All detected ion species with
enabled formula assignments in present samples are listed in Table S3.

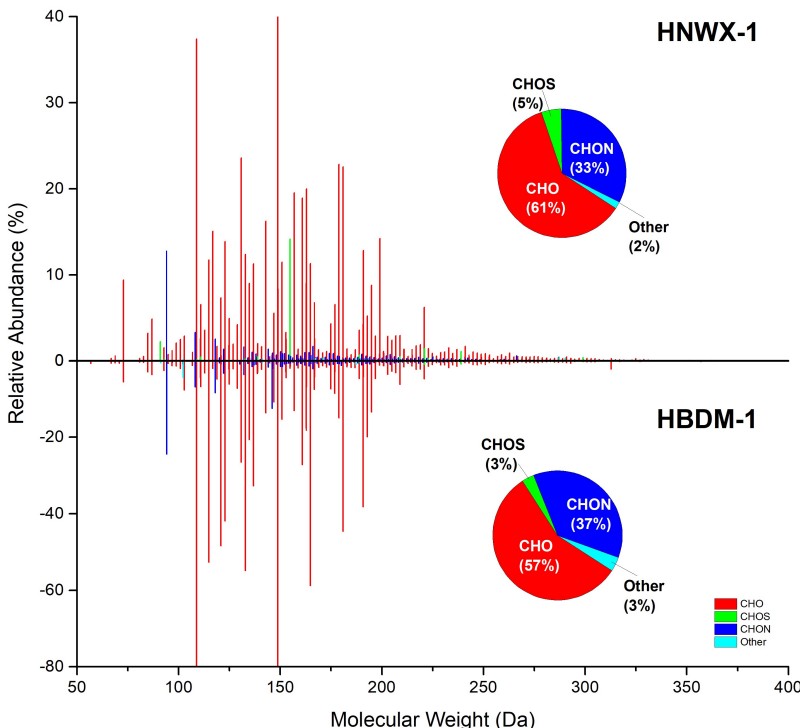


**Figure 2. Reconstructed mass spectra of HNWX-1 and HBDM-1 sample. The inset pie**
**charts show the number fraction of each class in the total assigned compounds.**

The negative ion mode is prone for the detection of molecules containing polar
functional groups (e.g., -OH and –COOH). It should be noted that the formula
numbers detected in the HRMS potentially contain multiple structural isomers;



therefore, the actual number of water-soluble organic species might be
underestimated. The additional LC/ESI-HRMS analysis operated in negative mode
confirmed a substantial number of ion masses (e.g., assigned CHO and CHON
compounds) containing more than one structural isomer, which could be observed at
different retention times (RTs) in chromatograms. Two representative groups of
extracted chromatograms for CHO ([$C_7H_5O_n$]$^-$, (n=2~4)) and CHON ([$C_7H_5O_nN$]$^-$,
(n=1~3)) compounds are shown in Figure S2 and S3, respectively, where increasing
the O or N atom number in a molecule might lead to more isomer peaks. However, it
should be noted that these LC-separated peaks might also include other unidentified
compounds that were outside of the elemental assignment considered in this study.
Additionally, low content and potential decomposition under the ionization can also
limit the detection of some high molecular weight species.
The interpretation of the complex organic mass spectra generated by high
resolution mass spectrometry can be simplified by plotting the hydrogen to carbon
ratio(H/C) against the oxygen to carbon ratio (O/C) for individual assigned atomic
formulas in form of the Van Krevelen (VK) diagram (e.g. Lin et al., 2012a;
Kourtchev et al., 2013). Figure 3a indicates a representative VK diagram of CHO
and CHON compounds derived from HBDM-1 sample. It can be clearly seen from
Figure 3a that the majority of CHO and CHON molecules are located at the region of
O/C≤1.0 and H/C≤2.0. In VK diagram, molecules with H/C≤1.0 and O/C≤0.5
are typical for aromatic species, while molecules with H/C≥1.5 and O/C≤0.5
would be associated with typical aliphatic compounds (Mazzoleni et al., 2012;



Kourtchev et al., 2014). The average double bond equivalent (DBE) showed relative
high values with 5.5 for CHO compounds and 6.1 for CHON compounds (Table S2),
suggesting that oxidized aromatic compounds were abundant in the present sample,
and their presence could partially account for the strong light-absorbing feature in
the near-UV region as observed in our previous study (Cai et al., 2018). Figure 3b
shows the distribution of molecular formulas with various DBE and indicates a large
number of molecular species with high unsaturation degree (DBE≥4).

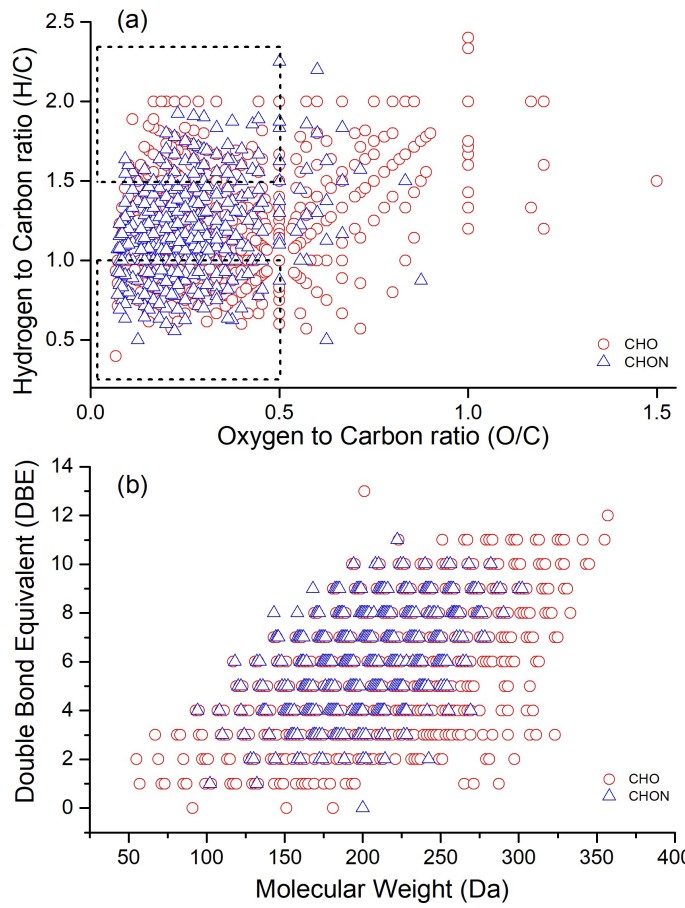


**Figure 3. VK diagram (a) and DBE vs. molecular weight (b) of CHO and CHON**
**compounds for one representative sample (HNDM-1).**





The average H/C and O/C ratios throughout the all extract samples were in the
ranges of 1.26-1.31 and 0.34-0.42 for CHO compounds, 1.19-1.23 and 0.28-0.29 for
CHON compounds (shown in Table S2), respectively. Although the ESI analysis
were performed in the negative ionization mode, the emerged O/C ratios exhibit
rather low values, which fall in the range of O/C ratios typical for biomass burning
organic aerosol derived from positive ionization mode (Aiken et al., 2008;
Kourtchev et al., 2016). Due to fresh emission and smaller aging effect, the present
O/C were obviously lower than the O/C of long-range transport biomass burning
aerosols (Zhang et al., 2018).
Carbon oxidation state ($OS_c$) was observed to increase with oxidation for
atmospheric organic aerosol and link strongly to aerosol volatility (Kroll et al., 2011).
$OS_c$ for each molecular formula can be calculated using the following equation:

$$OS_c = -\sum_i OS_i \frac{n_i}{n_c}$$

where $OS_i$ is the oxidation state associated with element $i$ and $n_i/n_C$ is the molar ratio
of element $i$ to carbon within the molecule (Kroll et al., 2011; Kourtchev et al., 2013).
Figure 4 shows an overlap in $OS_c$ plots of CHO compounds for two representative
samples (HNWX-1 and HBDM-2) derived from different sampling sites. It can be
seen that $OS_c$ ranges mainly from -1.5 to +1 with an average of 0.4. Consistent with
previous studies (Kroll et al., 2011; Kourtchev et al., 2016), the majority of molecules
with $OS_C<0$ (low oxidized organics) and carbon atoms ($n_C$) lower than 20 are
suggested to be associated with the primary organic aerosols emitted from biomass
burning. A minor fraction of molecular formulas with $OS_C \geqslant 0$ values might be



associated with semivolatile and low-volatility oxidized organic aerosols (Kroll et al.,

2011).

A similar trend of $OS_c$ values versus carbon number was obtained in previous

studies focused on the molecular composition of organic aerosols in urban area (Wang
et al., 2017) and at a road tunnel site (Tong et al., 2016), although the formulas of the
specific molecular products observed from different precursors in both studies are
quite different.

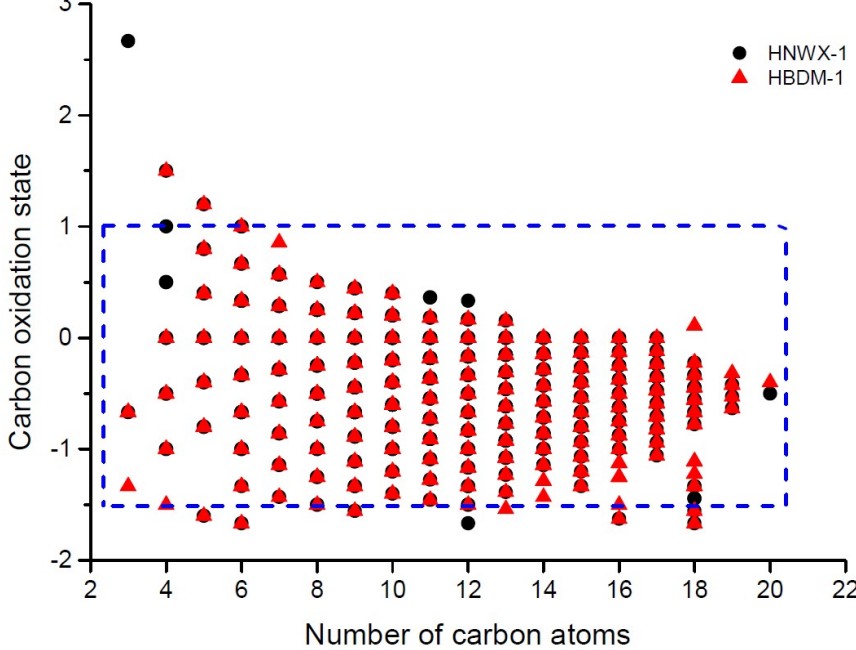


**Figure 4. Carbon oxidation state ($OS_C$) vs. number of carbon atoms in CHO molecules.**
**3.2 Photochemical oxidation of phenols under laboratory conditions**

Phenol and guaiacol were chosen as two representative model compounds derived

from biomass combustion. Two high resolution mass spectra of aqueous phenol and



guaiacol exposed to OH radicals for 4h, are shown in Figure S4. In Figure S4, 435
$C_xH_yO_z$ molecular formulas (m/z 90-500) were assigned for product ions of phenol
(with $C_3$-$C_{24}$), whereas 624 $C_xH_yO_z$ formulas (m/z 90-600) were assigned for product
ions of guaiacol (with $C_3$-$C_{27}$). The average H/C and O/C ratios were $0.79\pm0.28$ and
$0.52\pm0.23$ for phenol, and $0.88\pm0.24$ and $0.59\pm0.24$ for guaiacol, respectively.
Clearly, the photochemical processing induced by OH oxidation resulted in an
increase in O/C of product molecules relative to their precursors (O/C=0.17 for
phenol and O/C= 0.29 for guaiacol). Meanwhile, the average $OS_C$ of products for
phenol ($OS_C$ = -0.7) and guaiacol ($OS_C$= -0.6) photooxidation were +0.2 and +0.3,
respectively, showing an increase with oxidation. The later implies that potentially
the phenols and methoxyphenols might undergo photochemical aging and thus alter
the nature of primary organic aerosols (Huang et al., 2018).
The formation mechanisms of series of oxygenated products, e.g., phenolic
oligomers, hydroxylated phenolic species, ring-opening and highly oxygenated
compounds, are proposed in the literature (e.g. Sun et al., 2010; Chang and
Thompson, 2010, Yu et al., 2014). The OH-initiated reactions would result in
enhanced hydroxylation of the aromatic ring as well as to increased yields of
carboxylic acids and toxic dicarbonyl compounds (Sun et al., 2010; Yu et al., 2014;
Prasse et al., 2018). For example, some highly oxygenated $C_2$-$C_5$ aliphatic
compounds (e.g., $C_2H_2O_4$, $C_3H_4O_4$, $C_4H_6O_4$, and $C_5H_6O_5$) corresponding to
carboxylic acids (Yu et al., 2014) were clearly observed in the mass spectra of
present photochemical products. The presence of these oxygenated products not only



directly increased the degree of oxygenation in the bulk solution composition, but
also contributed to the variation of solution acidity. The pH measurements indicated
that the acidities ($[H^+]$) of the bulk solution increased by $(2.96\pm0.15)\times10^{-5}$ M and
$(4.26\pm0.16)\times10^{-5}$ M for phenol and guaiacol, respectively.

The oligomerization induced by photochemical transformation of phenolic

substances is an important formation pathway for the low-volatility, light-absorbing
compounds (Smith et al., 2016). Here, phenolic dimmers (i.e., $C_{12}H_{10}O_2$ for phenol
dimer and $C_{14}H_{14}O_4$ for guaiacol dimer) and higher oligomers (e.g., $C_{18}H_{14}O_3$ and
$C_{24}H_{18}O_4$ for phenol trimer and tetramer, $C_{21}H_{20}O_6$ for guaiacol trimer), as well as
their hydroxylated species were observed. The formation mechanism, can be
ascribed to C-O or C-C coupling of phenoxy radicals that were formed via
H-abstraction of the phenols or OH addition to the aromatic ring (Net et al., 2009,
Sun et al, 2010). The reaction at the para position or para-para coupling was more
likely to occur due to a higher probability of free electron to occur in this position
(Lavi et al, 2017) or a weaker steric hindrance in the para position. The extracted LC
diagrams of m/z 185.0608 and 245.0823 are shown in Figure 5a and Figure 6a,
respectively, where both ions involve dimers of phenol and guaiacol with several
structures, and/or other isomers. The presence of guaiacol dimer and syringol dimer
was previously observed in aerosol samples largely affected by wood combustion.
Based on the Aerosol Mass Spectrometer (AMS) analysis, these two dimers were
suggested as markers of biomass burning aerosols (Sun et al., 2010; Yu et al., 2014).
In the composition of present biomass burning aerosols, the phenolic dimers (m/z
185.0608 and 245.0823) were also observed in present mass spectra, but the
extracted LC diagrams shown in Figure 5b-c and Figure 6b-c indicate that these ions
contain multiple RT peaks. The same peaks with RT18.3 and 19.2 min which are
assumed to be the phenol dimers were observed during the photochemical
transformation of phenol (Figure 5a) and in the straw-burning samples (Figure 5b-c).
Meanwhile, the present particle extracts may also involve guaiacol dimer, since its
m/z 245.0823 has two LC peaks emerged at RT 17.7 and 19.5 min (Figure 6b-c)
same as the peaks identified during the photochemical transformation of guaiacol
(Figure 6a).

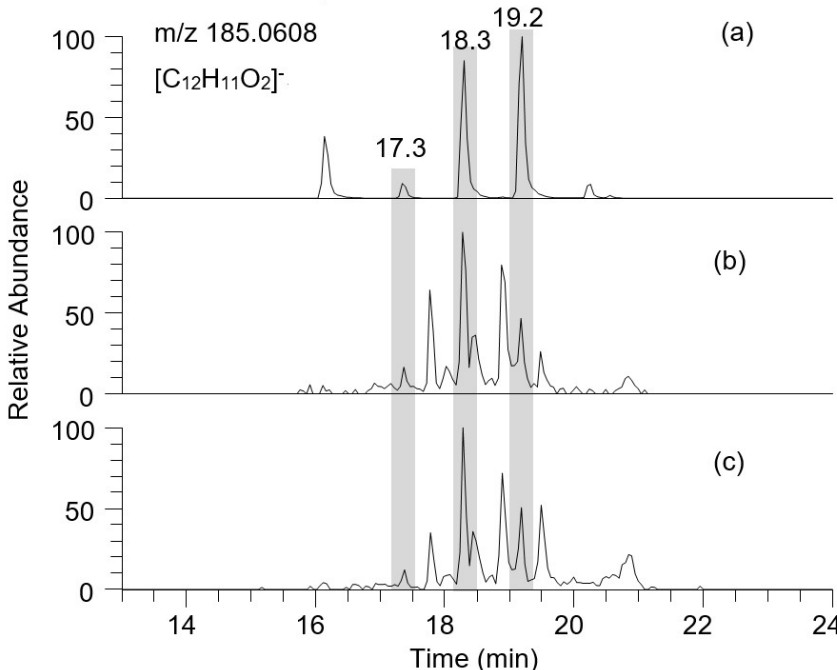


**Figure 5. Extracted LC chromatograms of m/z 185.0608 in (a) photochemical sample of**
**phenol, (b) HNWX-1, and (c) HBDM-2.**

Typical hydroxylated species such as, e.g., $C_2H_2O_4$, $C_6H_6O_2$, $C_7H_6O_3$, $C_7H_8O_3$,



were also found in the samples emerged from the photooxidation of both phenols
and the straw-burning samples. The comparison of the photooxidation products
stemmed from the phenols and the straw-burning samples revealed their significant
difference, pointing to the importance of studying real aerosol samples against the
laboratory model compounds. However, evaluating the model compounds as proxy
of real aerosol samples is always helpful as a reference. To this end, it is worth
noting that potentially other phenols and methoxyphenols (e.g., acetosyringone,
vanillin) that dissolve into cloud, fog droplets or aerosol liquid water can be
photochemically transformed and contribute to the SOA formation (Vione et al.,
2019, Zhou et al., 2019).

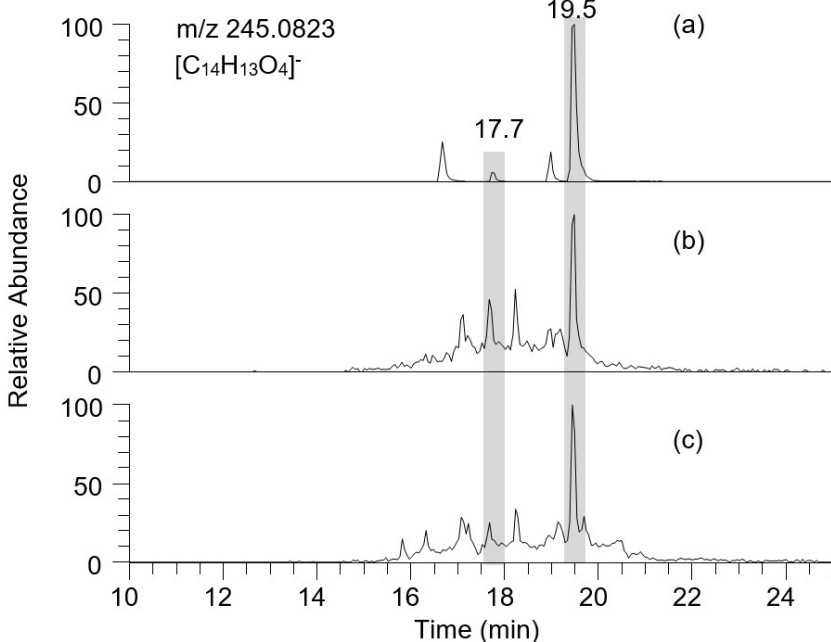


**389 Figure 6. Extracted LC chromatograms of m/z 245.0823 in (a) photochemical sample of**
**390 guaiacol, (b) HNWX-1, and (c) HBDM-2.**

**391 3.3 Photochemistry of aqueous extracts derived from straw burning aerosols**



Although the direct photolysis was performed on present straw burning samples in
presence of simulated sunlight irradiation without adding any oxidants, the
photooxidation process still occurred since the particle extracts were very likely to
include various oxidants, e.g., singlet molecular oxygen ($^1O_2$), peroxides, hydroxyl
radical (OH) or excited triplet state of organics produced under light excitation
(Anastasio et al., 1997; Vione et al., 2006; Net et al., 2009; Net et al., 2010; Bateman
et al., 2011; Rossignol et al., 2014; Smith et al., 2014; Gómez Alvarez et al., 2012).
In particular, the excited triplet state of aromatic carbonyls (e.g., 3,
4-dimethoxybenzaldehyde) was found to be more efficient than OH radical to
oxidize phenols and produce hydroxylated species (Smith et al., 2014., Yu et al.,
2014). This photosensitized reaction is likely to play an important role in the WSOC
evolution, due to high quantities of aromatic carbonyls present in the extracts of
biomass burning aerosols.
Although no available standards were utilized for absolute quantification, the
variation in peak abundance at unique retention times in the chromatogram could
reflect the extent of evolution of WSOC molecules, with accurate molecular weights.
The LC/ESI-HRMS monitors changes in the molecular features of a part of the
WSOC, i.e., photodegradation of low oxygenated compounds and formation of high
oxygenated compounds. Table 1 lists the CHO compounds for which the LC peak
intensities significantly increased and decreased after the 12-hour photolysis.
**3.3.1 Photodegradation of low oxygenated compounds and formation of highly**
**oxygenated compounds**



As shown in Table 1, ion masses assigned with high unsaturated and low
oxygenated species (O/C＜0.5) are prone to photodegradation, especially $C_7$-$C_9$
compounds (possible aromatic species), which intensity decreased by nearly one
order of magnitude. For example, for m/z 123.0450 ([$C_7H_7O_2$]⁻), as shown in Figure
7a, the peaks at RT 16.2 and 16.7 min in the LC chromatogram reduced in area by 95%
after the 12-h irradiation. Using a standard it was verified that both peaks did not
belong to guaiacol (peak at RT17.3 min), but they were also found within the
products of guaiacol photo-oxidation, suggesting that they might be isomers of
guaiacol or aromatic dihydric alcohol.
**Table 1. *M/Z* with significant changes upon 12-h photolysis analyzed by LC/ESI-HRMS.**

| Precursor (LC peak intensity decreases by ＞50%) | | | Product (LC peak intensity increases by ＞50%) | | |
|---|---|---|---|---|---|
| Retention time, min | Measured *m/z* | Molecular formula | Retention time, min | Measured *m/z* | Molecular formula |
| 16.2,16.7 | 123.04497 | $C_7H_8O_2$ | 1.9 | 59.01362 | $C_2H_4O_2$ |
| 13.9,14.5 | 129.05555 | $C_6H_{10}O_3$ | 1.8 | 72.99291 | $C_2H_2O_3$ |
| 14.6 | 131.07121 | $C_6H_{12}O_3$ | 2.1 | 73.02928 | $C_3H_6O_2$ |
| 14.6 | 133.02934 | $C_8H_6O_2$ | 1.8 | 75.00856 | $C_2H_4O_3$ |
| 15.9 | 135.04498 | $C_8H_8O_2$ | 2.4 | 85.02930 | $C_4H_6O_2$ |
| 13.7 | 137.02426 | $C_7H_6O_3$ | 1.9, 4.4 | 87.04496 | $C_4H_8O_2$ |
| 17.7 | 137.06063 | $C_8H_{10}O_2$ | 1.9 | 88.98785 | $C_2H_2O_4$ |
| 15.8 | 147.04504 | $C_9H_8O_2$ | 1.9 | 89.02427 | $C_3H_6O_3$ |
| 17.2 | 149.06062 | $C_9H_{10}O_2$ | 2.2 | 99.00857 | $C_4H_4O_3$ |
| 19.0 | 151.07634 | $C_9H_{12}O_2$ | 2.5 | 129.01917 | $C_5H_6O_4$ |
| 16.8 | 161.06068 | $C_{10}H_{10}O_2$ | 2.0 | 145.01407 | $C_5H_6O_5$ |
| 16.2 | 165.05559 | $C_9H_{10}O_3$ | 1.9 | 147.02971 | $C_5H_8O_5$ |
| 14.9 | 167.07129 | $C_9H_{12}O_3$ | 14.9 | 155.03482 | $C_7H_8O_4$ |
| 15.1 | 181.05048 | $C_9H_{10}O_4$ | 15.1 | 169.01411 | $C_7H_6O_5$ |
| 17.3 | 191.03498 | $C_{10}H_8O_4$ | 16.4 | 183.02980 | $C_8H_8O_5$ |
| 16.2 | 195.06622 | $C_{10}H_{12}O_4$ | | | |
| 18.6 | 207.06635 | $C_{11}H_{12}O_4$ | | | |






The phenolic dimers ($C_{12}H_{10}O_2$ and $C_{14}H_{14}O_4$) as described above also exhibited a
decreasing tendency with almost complete disappearance after 12h direct photolysis.
Other species with relatively high MW ($\geq$200Da) were also observed to be
decomposed, including m/z 251.0564 ($[C_{12}H_{11}O_6]^-$), 313.0724 ($[C_{17}H_{13}O_6]^-$), and
329.0674 ($[C_{17}H_{13}O_7]^-$) (Figure S5), although their initial abundance was not very
high.
On the other hand, the solution acidity ($[H^+]$) of the particle extracts increase after
the 12-hour photolysis, similar to the observation on the photo-oxidation of phenols
(section 3.2) that resulted in the formation of oxygenated species. The solution
acidity ($[H^+]$) normalized by WSOC concentration ($[OC_{ws}]$) was increased with a
variation of $\Delta[H^+]/[OC_{ws}]=(3.8\pm0.8)\times10^{-7}$ mol mgC$^{-1}$, suggesting the formation of
new acidic substances.
The photochemical processing has led to an increased formation of low MW
compounds (e.g., $C_2$-$C_5$ species), with a relatively high O/C ratio. For example, the
$C_2$ compounds, including $[C_2H_1O_3]^-$, $[C_2H_3O_3]^-$, $[C_2H_3O_2]^-$, and $[C_2H_1O_4]^-$ (Figure
S6), which may correspond to glyoxylic acid, glycolic acid, acetic acid, and oxalic
acid, respectively, were likely to be formed via oxidation pathway of several
water-soluble molecules with photochemical reactivity (e.g., glyoxal (Carlton et al.,
2007; Lim et al., 2010), methylglyoxal (Altieri et al., 2008; Lim et al., 2010),
pyruvic acid (e.g. Grgic et al., 2010; Griffith et al., 2013; Reed Harris et al., 2014;
Rapf et al., 2017; Eugene and Guzman, 2017, Mekic et al., 2018), phenols (Sun et al.,
2010), etc). The presence of these highly oxygenated compounds that possibly

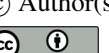

contain acidic groups (e.g., –COOH and –OH) undoubtedly contributed to the
increase of the solution acidity. Higher levels of other highly oxygenated species
such as $[C_3H_5O_3]^-$, $[C_4H_7O_2]^-$, $[C_5H_5O_5]^-$ and $[C_5H_7O_5]^-$ were also observed (Figure
S7).

To identify the impact of photolysis on the evolution of specific WSOC, the ions

of $[C_7H_7O_n]^-$ in the HBDM-1 sample with significant variation were chosen as
representative cases for description. The relative intensity of $[C_7H_7O_2]^-$ and
$[C_7H_7O_3]^-$ decreased dramatically, while the intensities of $[C_7H_7O_4]^-$, $[C_7H_7O_5]^-$ and
$[C_7H_7O_6]^-$ increased with photolysis (Figure 7 just shown the variation of $[C_7H_7O_2]^-$
and $[C_7H_7O_4]^-$). It seems reasonable that the possible hydroxylation of $[C_7H_7O_2]^-$ and
$[C_7H_7O_3]^-$ might contribute to the formation of $[C_7H_7O_5]^-$ and $[C_7H_7O_6]^-$. Although
we could not verify this hypothesis, the formed oxidized species undoubtedly have a
high O/C ratio which highlights the possibility of this reaction pathway.

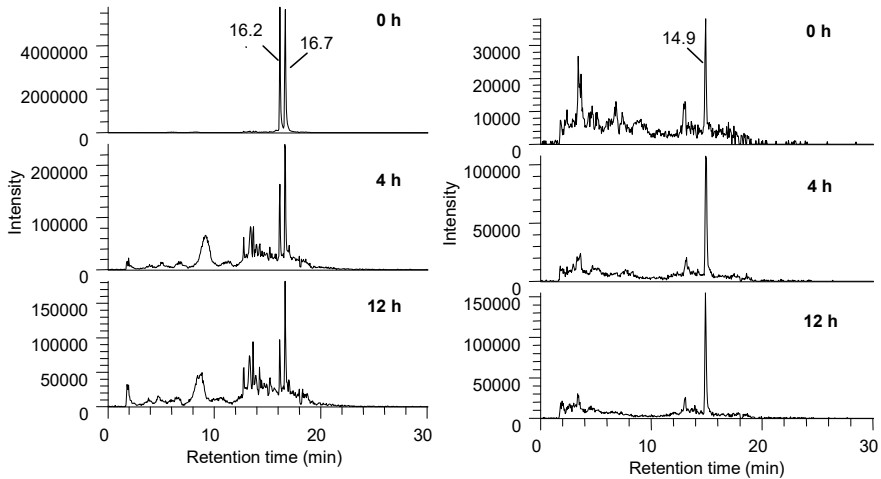


**Figure7. Extracted LC chromatograms from HBDM-2 of (a) $[C_7H_7O_2]^-$ and (b) $[C_7H_7O_4]^-$ at**
**different photolytic stages of 0, 4, and 12 h.**



### 3.3.2 Effect of photolytic processing on other WSOC

Some of the detected organic species exhibit a good photochemical stability, as their relative intensities only slightly decreased (<10%) after 12h light irradiation. The m/z 161.0454 ($[C_6H_9O_5]^-$) presented two prominent peaks at RT1.9 and 2.4 min (Figure S8). The peak at RT 2.4 min was suggested to be levoglucosan, a typical tracer of biomass burning aerosols (Hu et al., 2013). Its presence in the sample was further confirmed with a standard compound. The relatively good photochemical stability was also observed for some $C_6$ homolog compounds, such as $[C_6H_7O_6]^-$, $[C_6H_9O_6]^-$, and $[C_6H_{11}O_6]^-$. Some other oxygenated species, such as $[C_3H_3O_3]^-$, $[C_4H_5O_4]^-$, $[C_3H_3O_4]^-$, and $[C_4H_5O_5]^-$ remained relatively stable, as well.

Regarding the CHON compounds, only small variation of the chromatogram peaks, was observed for most of the detected species. In particular, several species with low O/C decreased by less than 30%, e.g., m/z 94.0297 ($[C_5H_4ON]^-$, RT 7.1 min), and 120.0453 ($[C_7H_6ON]^-$, RT12.2 min). Some compounds were photochemically very stable as the variation of their peak intensities was less than 10 % upon light irradiation of the samples, e.g., m/z 118.0297 ($[C_7H_4ON]^-$, RT16.6 and 17.1 min), 146.0246 ($[C_8H_4O_2N]^-$, RT14.4 min), and 190.0510 ($[C_{10}H_8O_3N]^-$, RT17.8 min). However, the intensities of the ion masses with relatively higher degree of oxygenation was found to increase substantially (>50%), e.g., m/z 162.0195 ($[C_8H_4O_3N]^-$, RT 17.2 min), 198.0408 ($[C_8H_8O_5N]^-$, RT 18.0 min), and 242.1763 ($[C_{13}H_{24}O_3N]^-$, RT 17.9 min).

Another intriguing finding was that different structural isomers with the same





molecular mass might have exhibited different fates upon prolonged light irradiation
of the samples. For example, the intensity of the peak at m/z 165.0405 ($[C_5H_9O_6]^-$)
decreased when it was eluted at 4.9 min, but increased at RT 1.8 min, with the
irradiation time (Figure S9). A simultaneous degradation and formation among
isomers of some CHON ion masses upon prolonged light irradiation, was also
observed, as was the case for the CHO compounds. For example, the m/z 108.0453
assigned to $[C_6H_6ON]^-$, might include hydroxy and amino groups on the phenyl ring
to present three possible isomers (Figure S10). During photolytic processing, the
intensity of the peak at RT 3.2 min increased dramatically, while there was a clear
decreasing tendency of the peak intensity at RT 5.5 and 12.5 min, which was
suggestive of possible isomerization among these isomers. Other ion masses that
exhibited possible isomerization included m/z 122.0610 ($[C_7H_8ON]^-$), 132.0454
($[C_8H_6ON]^-$),    134.0245    ($[C_7H_4O_2N]^-$),    136.0403    ($[C_7H_6O_2N]^-$),    138.0559
($[C_7H_8O_2N]^-$), 144.0453 ($[C_9H_6ON]^-$), and 152.0352 ($[C_7H_6O_3N]^-$).
**3.3.3 Effect of photolytic processing on mass spectral features of WSOC**

Since the LC method just separated a fraction of polar compounds, we tentatively

utilized the change of HRMS to gain more comprehensive information about the
WSOC evolution. We compared the time-profile (0, 4, and 12h) mass spectra with
each other, based on the assumption of same interference from inorganic species, and
the good reproducibility and stability for orbitrap MS operated under the same
instrumental parameter (the RSD of TIC intensity within 5%). It is well known that
ESI mass spectral abundances are influenced by the solution composition,





concentration of analytes and instrumental factors (Bateman et al., 2011); hence, it is
quite challenging to directly quantify the absolute concentration levels of the
complex mixtures. Despite that, the photochemical degradation of WSOC
compounds and corresponding formation of organic compounds can be well
described by the variation of signal intensity from mass spectrometry. The average
O/C and H/C ratios for CHO compounds were from $0.38\pm0.02$ to $0.44\pm0.02$ and
$1.24\pm0.03$ to $1.26\pm0.01$, respectively, as the irradiation time extended up to 12h.
The comparison of these time-profile mass spectra indicates that the 12-hour
photolysis resulted in a significant reduction of $28\pm11\%$ in the total ion abundance
(S/N). Since the photolysis induced changes in abundance for most of the CHO
compounds, we also calculated the intensity (S/N)-weighted average O/C (O/C$_w$) and
H/C (H/C$_w$) (Bateman et al., 2011; Romonosky et al., 2015) with values ranging
from $0.45\pm0.03$ to $0.53\pm0.06$ and from $1.32\pm0.09$ to $1.40\pm0.11$, respectively.
Both average O/C ratios with and without intensity-weighted showed an increased
tendency that indicated an elevation in the degree of oxygenation of bulk extract
composition, consistent with the LC observation, i.e. formation of highly oxygenated
species and the consumption of low oxygenated compounds. This result bears
similarity with previous observation using ESI mass spectrometry on characterizing
photochemical transformations of d-limonene in the aqueous phase as a source of
SOA (Bateman et al., 2011).
**4 CONCLUSIONS**
This study was focused on the effect of direct photolysis on the molecular



composition of actual WSOC extracted from fresh straw-burning aerosol. The
photooxidation of phenols in the aqueous phase under laboratory conditions
indicates that the phenols in real biomass burning WSOC would likely have potential
to experience the similar evolution to form various oxygenated compounds under
relevant cloudwater condition. Because the extract composition was very complex,
the high-resolution mass spectrometers used in this study (ESI-HRMS and
LC/ESI-HRMS), although advanced still had limitations in monitoring the
modification of molecular composition, especially for determining the potential
formation of compounds present at low concentrations or compounds that were
poorly ionized. However, a series of polar molecules were identified that changed
their molecular composition via photochemical evolution. In particular, the
degradation of low oxygenated compounds with strong photochemical reactivity and
the formation of high oxygenated compounds might directly result in an increasing
O/C ratio in WSOC. This finding indicates that the water soluble organic fraction of
fresh combustion-derived aerosols have the potential to form more oxidized organic
matter, which might partly account for the highly oxygenated nature of atmospheric
organic aerosols. High MW ion masses (MW≥300Da) typical for oligomers were
also found in the degradation products (even some with low abundance). Some CHO
and CHON species exhibited no significant losses (<10%), and displayed good
photochemical stability, which indicates that they may also be potential candidates of
tracers of biomass burning aerosols.
**AUTHOR CONTRIBUTION**



Jing Cai and Zhiqiang Yu designed the experiments, and Jing Cai and Xiangying
Zeng carried them out. Guorui Zhi provided the straw-burning aerosol samples,
Sasho Gligorovski helped perform the analysis of light irradiation. Guoying Sheng,
Xinming Wang and Ping'an Peng provided some technical consultations about
organic chemistry. Jing Cai prepared the manuscript with contributions from all
co-authors.
**ACKNOWLEDGMENTS**
This study was financially supported by the National Key Technology Research and
Development Program of the Ministry of Science and Technology of China
(2014BAC22B04), the National Natural Science Funds of China (41225013,
41530641, and 41373131) and the Science and Technology Project of Guangdong
Province, China (2014B030301060).

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
