# Peer review of "Molecular Composition and Photochemical Evolution of Water Soluble Organic Carbon (WSOC) Extracted from Field Biomass Burning Aerosols using High Resolution Mass Spectrometry"

_Atmospheric Chemistry and Physics, 2019_

## Referee Comment (RC1) · Anonymous Referee #1 · 27 Oct 2019

General comments:

In this paper, the authors use direct high resolution mass spectrometry and liquid chromatography with high resolution mass spectrometry to study the aging of a water soluble organic carbon mixture from biomass burning aerosol. They also study the photochemistry of phenol and guaiacol in the aqueous phase, for comparison to their ambient results. This work provides an interesting combination of experimental methods and laboratory analyses to probe the composition and evolution of water soluble organic compounds at the molecular-level. However, prior to acceptance, I recommend

the following clarifications in the presentation and discussion of the data, as well as the following grammatical corrections.

Specific comments:

In general, I think the structure of this paper could be improved. It is not immediately clear whether the data discussed in section 3.1 (Mass spectral characteristics of WSOC in biomass burning particulate) are from fresh particles. It seems to be discussing fresh particles, based on the titles of the other sections, but this should be made clearer in the text or in the section title. Section 3.2 (Photochemical oxidation of phenols under laboratory conditions) shows interesting results, but these results could be linked more explicitly to the ambient data presented in the paper. There is a short discussion of a comparison between laboratory and ambient data (lines 379-387) and other discussions in the following section (3.3), but as a reader, I find this information hard to keep track of. Perhaps the paper could benefit from a specific section for lab-ambient intercomparisons. Section 3.3.2 seems to be discussing photochemical stability—consider labeling more clearly as such. Section 3.3.3 seems to be discussing changes in composition as a result of different aging times—consider labeling more clearly as such.

The authors use straw burning aerosol in this study. How representative is straw as a fuel in the particular region the authors are studying? How representative is this fuel more generally? This should be addressed—there are lots of different types of fuel that are burned and the choice of straw should be put into appropriate context.

This study uses negative mode ionization only: how might this skew the types of compounds/compound classes identified? This should be discussed. Negative mode will ionize compounds that can be readily deprotonated, but what about compounds that are not easily deprotonated and may show up preferentially in positive mode ionization (e.g. compounds like amines)? How representative are the data in encapsulating mixture-wide characteristics if positive mode is not used?

Line 211: CHS compounds do not ionize well with electrospray ionization, which may explain why they are not detected with much prevalence here. This should be discussed.

Are the molecular formulas with the lowest ppm mass difference selected here? Are there any other QC/QA methods you use, like ensuring H/C ratios are reasonable or checking for non-integer DBEs in neutral formulas?

In general, I find the methods a bit challenging to follow. There are lots of interesting sampling and analytical methods used here and the differences between them for accurately interpreting the results are important. Perhaps the authors could include a summary table or flow chart of the different types of sample collection methods and data analysis methods for field and ambient samples.

Line 228-231: Are these percentages by occurrence? By abundance? If weighted by abundance, how is ionization efficiency accounted for?

In Figures 3 and 4, what do the boxes around the data mean? This should be mentioned in the figure caption.

In Figures 2-4, the authors show 1-2 representative samples. How do we know that these samples are truly representative? The authors should consider showing the rest of the data in the SI and highlighting how similar the data are, or perhaps should consider finding a way to show averages across all samples in main text figures.

Line 325-327: I agree that there is an increase in the average O/C of product compounds relative to their precursors, but this is to be expected. There are many products formed during phenol and guaiacol photochemistry, some of which probably have higher O/C than the precursor compounds and some of which may fragment and have lower O/C. While the average O/C of the product mixture increases, it should be acknowledged that this is an average and that many different product compounds are formed.

Line 343-345: What were the starting pH values and are these changes statistically significant?

Figures 5 and 6 don't show drastically different information, they can probably be combined.

Lines 511-526: It should be clearer which time points you are referring to for these O/C and H/C ratios (t=0? T=4? T=12?).

Line 547: Past work that has discussed the relatively short lifetime of CHON compounds like nitrophenols (order of hours, depending on conditions). In this manuscript, the authors mention that CHON compounds tended to exhibit good stability. What do you estimate the atmospheric lifetime of CHON compounds observed in your analysis to be and how does this compare to past literature?

Technical corrections:

The quality of the writing in this manuscript should be improved upon prior to publication. These are some suggested edits, but in general the authors should carefully review the language in their manuscript.

Abstract

-Line 34: remove comma between "precursor" and "were"

-Line 35: remove comma after "both"

-Line 39: "of" instead of "on"

-Line 41: "extracts" instead of "extract"

-Line 47: "has" instead of "have"

Introduction

- Line 53: space before "WSOC"

- Line 61: space before "Although"

- Line 66-67: for clarity, please define functional groups in words first before using abbreviated descriptions

- Line 68: "carbonyls" instead of "carbonyl"

- Line 71: could be clarified, what do you mean by "to affect aerosol evolution processes"?

- Line 82: consider removing the word "emerged", its meaning is unclear here

- Line 84: "environments" instead of "environment"

- Line 109-112: this sentence needs re-structuring for clarity: be more explicit and clear about why LC will help relieve ion suppression and how it will help identify ions (both are possible because of differences in LC retention time between compounds, but this should be made more obvious, for readers who do not frequently use LC techniques)

- Line 118: "were" instead of "was"

- Line 120: the word "reference" does not seem to be what you mean to say here, what about changing to "for comparison"?

Methods - Line 158: What is this ratio in your extraction vs. in actual cloud water?

- Line 160: "In the experimental section of phenol photochemistry" is confusing, perhaps re-phrase to "To study phenol photochemistry"

- Line 160: "solution" should be "solutions"

- Line 162: remove comma after "H2O2"

- Line 166-169: should be re-phrased, also I imagine you do not mean that you are looking for "biomarkers" but instead "tracer compounds"?

- Line 171: define "ESI" the first time you use it

- Line 176: Orbitrap should be capitalized consistently throughout the text

- Line 185: the plural of "spectrum" is "spectra"

- Line 208: please write out the full name of the "nitrogen rule"

Results and Discussion - Line 215: title seems incomplete, missing the word "matter"

- Line 216: "2.5" should be written as a subscript here and throughout the text (i.e. PM2.5)

- Line 224: "spectrum" should be "spectra" if the authors meant it to be plural, also what is meant by "abstracted blank"? I assume this should say "subtracted"

- Line 234-235: this sentence is incomplete

- Line 265: this sentence should be clarified: what is meant by "low content"? Low mass loading?

- Line 290: remove "emerged"

- Line 320: remove comma after "4h"

- Line 328: add "after" before the word "photooxidation"

- Line 321-331: should be clarified, if phenols and methoxyphenols are undergoing photochemical aging, how are they impacting POA?

- Line 351: remove comma after "mechanism"

- Line 357: for clarity, consider using "chromatograms" instead of "diagrams", and propagate through text

- Line 378: remove "emerged"

- Line 380: remove "stemmed"

- Line 505: "parameter" should be plural

- Line 520: should be re-phrased for clarity, what is meant by "increased tendency", what exactly is "consistent with LC observations"?

---

## Referee Comment (RC2) · Anonymous Referee #2 · 22 Dec 2019

This is an interesting manuscript that describes photo-oxidation of both representative "real" biomass burning organic aerosol (straw) as well as a simpler surrogate containing

I find the paper hard to follow. This is in part because the written English, while passable, is imperfect. However, the main issue is that there is no clear story beyond "oxidation of WSOC increases the oxidation state" (which is almost guaranteed) and "even a model system from a single fuel is very complex".

[Figure]

As far as I can tell a manuscript with a more clearly articulated story would be appropriate for ACP, but I am also on the edge of the subject area, and so a reviewer with greater experience with high-resolution GC methods might be more appropriate.

Some general comments:

The "/" in "O/C" really means ratio, so "O/C ratio" is redundant. I suggest writing "the oxygen to carbon ratio (O/C)" once and then omitting "ratio" when subsequently using the abbreviation. There is room in the literature for complex analyses of complex systems and we can not always demand an incredibly simple story, but the paper could still benefit from a major re-write to pull the most important themes to the surface.

I do not believe it is appropriate to end the abstract with "accounting for the highly oxygenated nature". Perhaps "contributing to" is warranted but the implication of the current wording is that the contribution dominates, and that has not been demonstrated here.

Specific comments:

Line 252 "is prone for" is not quite right. "is most sensitive to molecules containing polar ..."?

Line 287 "the all extract samples" ???? Either the authors actually mean "the all-extract samples" or they may mean "all of the extracted samples". Clarify.

Line 290 "the emerged O/C ratios" could be "the measured O/C values" ("values" is appropriate after O/C in my opinion).

Line 336 "as well as to increased" clashes with the subject "would result in" before, so "to" should be "in".

Line 347 "pathway for the low-volatility" strike "the".

Line 532 "experience the similar" again strike "the".

Line 543 "aerosols have the potential" the subject is "fraction" so should be "has".

Line 544 "partly account for" is better than the abstract but "contribute to" would be best in both places.

---

## Author Comment (AC1) · 2 Feb 2020

We would like to thank the reviewer for the constructive comments. We really appreciate these comments as they will surely lead to improved manuscript. Below are the point by point the answers to the reviewer comments. General comments: In this paper, the authors use direct high resolution mass spectrometry and liquid chromatography with high resolution mass spectrometry to study the aging of a water soluble organic carbon mixture from biomass burning aerosol. They also study the photochemistry of phenol and guaiacol in the aqueous phase, for comparison to their ambient results.

[Figure]

This work provides an interesting combination of experimental methods and laboratory analyses to probe the composition and evolution of water soluble organic compounds at the molecular-level. However, prior to acceptance, I recommend the following clarifications in the presentation and discussion of the data, as well as the following grammatical corrections. Thanks for the positive comments on our submission. We would like to further improve our manuscript following the concrete suggestions.

Specific comments: In general, I think the structure of this paper could be improved. It is not immediately clear whether the data discussed in section 3.1 (Mass spectral characteristics of WSOC in biomass burning particulate) are from fresh particles. It seems to be discussing fresh particles, based on the titles of the other sections, but this should be made clearer in the text or in the section title. Here the data discussed in section 3.1 are combined to present field-collected aerosol samples, e.g., wheat straw burning aerosol (WSBA) samples before photolysis. We have re-phrased it in section 2.1 and 3.1, and deleted the "fresh" in the text.

Section 3.2 (Photochemical oxidation of phenols under laboratory conditions) shows interesting results, but these results could be linked more explicitly to the ambient data presented in the paper. We have added a new section (i.e., section 3.3: Comparison of the photochemical products of phenolic compounds and the CHO composition in WSOC extracts from WSBA samples) to discuss the data comparison (including O/C, H/C and OSc) of WSBA samples and photochemical product samples (see lines 387-403).

There is a short discussion of a comparison between laboratory and ambient data (lines 379-387) and other discussions in the following section (3.3), but as a reader, I find this information hard to keep track of. Perhaps the paper could benefit from a specific section for lab ambient intercomparisons. We added a new specific section for inter-comparisons between lab and ambient samples (see section 3.3 in revised manuscript).

Section 3.3.2 seems to be discussing photochemical stabilityâËŸAËĞ consider labeling more clearly as such. The section title is changed as "Presentation of photochemically stable organic species"

Section 3.3.3 seems to be discussing changes in composition as a result of different aging timesâËŸAËĞ Tconsider labeling more clearly as such. The section title is changed as section 3.4.3 "Comparison of time-profile mass spectra of CHO composition in WSOC extracts from WSBA samples" (line 541).

The authors use straw burning aerosol in this study. How representative is straw as a fuel in the particular region the authors are studying? How representative is this fuel more generally? This should be addressed.âËŸAËĞ Tthere are lots of different types of fuel that are burned and the choice of straw should be put into appropriate context. According to Chinese government statistics (http://www.moa. gov.cn), there were 5.70 million acre and 14.2 million acre of wheat crop in Hebei province and Henan province in summer of 2019, respectively. Our sampling region was in Wenxian, in Henan province and Daming in Hebei province, where the wheat are the main crops in summer. To facilitate subsequent planting and management, a large amount of wheat straw was burned directly by farmers in the fields during the harvest season in 2013. This was not controlled burning, but an illegal open burning with random, unordered and uncontrollable features. We have properly addressed this question in section 2.1 (line 131-136).

This study uses negative mode ionization only: how might this skew the types of compounds/compound classes identified? This should be discussed. Negative mode will ionize compounds that can be readily deprotonated, but what about compounds that are not easily deprotonated and may show up preferentially in positive mode ionization (e.g. compounds like amines)? How representative are the data in encapsulating mixture-wide characteristics if positive mode is not used? Using negative mode ionization can identify those compounds that can be readily deprotonated, and our focus was on those compounds with common formulas including C, H, O, N and S atoms

(see section 2.5). It is true that a large amounts of not easily deprotonated compounds will show up preferentially in positive mode ionization (e.g. compounds like aldehydes, ketones, esters, amines), and they may also have different ionized forms (e.g., [M+H]+, [M+Na]+, [M+K]+. . .) that makes them hard to be identified. Furthermore, compounds containing polyfunctional groups cannot be also easily identified in negative mode, but rather in positive mode. We have mentioned the limitation in the text (line 279-283). The ionization method restricts the ability in encapsulating wide-mixture- characteristics of aerosols, because polar WSOC occupy only a fraction of aerosol composition, and uncertainties in those compounds (e.g., high molecular-weight matters) are out of detected range. Here we tentatively examine the molecular characteristics under our method limitation and cannot estimate the representation of identified compounds, especially lacking the data from positive mode detection.

Line 211: CHS compounds do not ionize well with electrospray ionization, which may explain why they are not detected with much prevalence here. This should be discussed.14 We have mentioned it at line 276-278.

Are the molecular formulas with the lowest ppm mass difference selected here? Are there any other QC/QA methods you use, like ensuring H/C ratios are reasonable or checking for non-integer DBEs in neutral formulas? Here the assignment of molecular formulas were based on lowest ppm mass difference between the measured and the theoretical ion (see section 2.5). The formula assignment was based on the measured mass, before determination, the Orbitrap analyzer was externally calibrated for mass accuracy using Thermo Scientific Pierce LTQ Velos ESI calibration solution (composed of m/z 265 sodium dodecyl sulfate, m/z 514 iodine sulfonic acid sodium and m/z 1079-1979 Ultramark polymers). We excluded those neutral formulas with non-integer DBEs as they were not common compounds that might be ionized with unreasonable H/C or included other elements.

In general, I find the methods a bit challenging to follow. There are lots of interesting sampling and analytical methods used here and the differences between them for accurately interpreting the results are important. Perhaps the authors could include a summary table or flow chart of the different types of sample collection methods and data analysis methods for field and ambient samples. We have supplemented the sampling and analytical methods in section 2.1: Line 131-149: Field campaign. To facilitate subsequent planting and management, a large amount of wheat straw is burned directly in the fields during the harvest season. The sampling fields were located in the north China plain, surrounded by wheat farm and far away from downtown. The smoke from open burning of wheat straw which enveloped the farm region was the main source of sampling particulate matter. (These fire events were not controlled burning process, but an illegal open burning with random, unordered and uncontrollable features. Because these events randomly occurred, the planning prior to events was not possible; hence, the samples reflect the real field situation.) Line 152-157: Extract preparation. Line 167-169 and 174-179: Photolytic experiment.

Line 228-231: Are these percentages by occurrence? By abundance? If weighted by abundance, how is ionization efficiency accounted for? By occurrence, meaning assigned molecular formula species (see line 259: In the amount of assigned formulas...), no ion intensity.

In Figures 3 and 4, what do the boxes around the data mean? This should be mentioned in the figure caption. We labeled the two areas with A and B, and added a sentence "Areas A and B are tentatively attributed to aliphatic and aromatic molecules, respectively.", see Figure 2 (in section 3.1) that replaces previous Figure 3. The previous Figure 4 now is replaced by Figure 3 (in section 3.1), and the dashed area in previous Figure 4 was deleted.

In Figures 2-4, the authors show 1-2 representative samples. How do we know that these samples are truly representative? The authors should consider showing the rest of the data in the SI and highlighting how similar the data are, or perhaps should consider finding a way to show averages across all samples in main text figures. We have re-arranged these figures. In the revised version, Figure 2 includes a reconstructed

mass spectrum and VK diagram of one representative sample, and other three samples are shown in Figure S2. The previous Figure 3 is deleted, instead of OSc including three samples (including laboratory samples). Other three field samples are shown in Figure S5.

Line 325-327: I agree that there is an increase in the average O/C of product compounds relative to their precursors, but this is to be expected. There are many products formed during phenol and guaiacol photochemistry, some of which probably have higher O/C than the precursor compounds and some of which may fragment and have lower O/C. While the average O/C of the product mixture increases, it should be acknowledged that this is an average and that many different product compounds are formed. Reviewer is right. Here we use "average O/C" instead of "O/C" (see line 358).

Line 343-345: What were the starting pH values and are these changes statistically significant? As described in section 2.3 (lines 189), the starting pH values were adjusted to 5, and these changes are statistically significant with T-test ($p < 0.05$). We have rephrased the sentence (see line 373).

Figures 5 and 6 don't show drastically different information, they can probably be combined. We have combined these two figures, see Figure 4 (in section 3.4) in the revised manuscript. Lines 511-526: It should be clearer which time points you are referring to for these O/C and H/C ratios (t=0? T=4? T=12?). T=0, as the irradiation time extended from 0 to 12h (see at line 556).

Line 547: Past work that has discussed the relatively short lifetime of CHON compounds like nitrophenols (order of hours, depending on conditions). In this manuscript, the authors mention that CHON compounds tended to exhibit good stability. What do you estimate the atmospheric lifetime of CHON compounds observed in your analysis to be and how does this compare to past literature? It is true that the lifetime of compounds depend on reaction conditions. Here some CHON compounds seemed to exhibit good stability, which might be caused by a limited reaction due to the low

mass content or light-shielding effect caused by other light-absorbing matter. We have mentioned it at line 523-525.

Technical corrections: The quality of the writing in this manuscript should be improved upon prior to publication. These are some suggested edits, but in general the authors should carefully review the language in their manuscript. We have edited carefully the language in the revised manuscript.

Abstract. -Line 34: remove comma between "precursor" and "were" Done (line 34).

-Line 35: remove comma after "both" Done (line 39).

-Line 39: "of" instead of "on" Done (line 43).

-Line 41: "extracts" instead of "extract" Done (line 46).

-Line 47: "has" instead of "have" Done (line 51).

Introduction - Line 53: space before "WSOC" Done (line 57).

- Line 61: space before "Although" Done (line 65).

- Line 66-67: for clarity, please define functional groups in words first before using abbreviated descriptions. Done (line 70).

- Line 68: "carbonyls" instead of "carbonyl". Done (line 72).

- Line 71: could be clarified, what do you mean by "to affect aerosol evolution processes"? We use "to generate low-volatility species" to replace "to affect aerosol evolution processes" (see line 75).

- Line 82: consider removing the word "emerged", its meaning is unclear here. The word "emerged" is deleted (line 84).

- Line 84: "environments" instead of "environment" Done (line 86).

- Line 109-112: this sentence needs re-structuring for clarity: be more explicit and clear

about why LC will help relieve ion suppression and how it will help identify ions (both are possible because of differences in LC retention time between compounds, but this should be made more obvious, for readers who do not frequently use LC techniques) "HRMS coupled with LC might be another complementary powerful tool for relieving ion suppression due to its abilities to separate and analyze different kind of compounds with differences in LC retention time."(see line 112-114).

- Line 118: "were" instead of "was" Done, we have rephrased the sentence (see line 124-128).

- Line 120: the word "reference" does not seem to be what you mean to say here, what about changing to "for comparison"? We have rephrased the sentence (see line 124-128), and the words of "for comparison" were deleted.

Methods - Line 158: What is this ratio in your extraction vs. in actual cloud water? Cai et al (2018), and Li et al (2017) reported that the formation of cloud droplets is associated with ambient liquid water content (LWC) values and PM 2.5 concentrations. In their observation at Mountain Tai in China, the LWC and PM2.5 concentrations during the cloud events ranged widely from 0.01 to 0.39 gm$-3$ and from 11.1 to 173.3 $\mu$g m$-3$, respectively. The high LWC could facilitate the formation of larger cloud droplets, whereas higher PM2.5 levels resulted in higher concentrations of water-soluble ions. The calculated ratio of LWC/ PM 2.5 mass from Li et al (2017) was in a wide range from $1.4\times102$ to $1.6\times104$, which is comparable with the ratios of extracted water mass/ PM mass (MH2O/ PM2.5) ranging from $1.8\times103$ to $3.4\times104$ in the present study, indicating that the aqueous extract from this study is similar with the cloud water. Please see the lines 180-185 of the revised version.

- Line 160: "In the experimental section of phenol photochemistry" is confusing, perhaps re-phrase to "To study phenol photochemistry" We have deleted it (line 187).

- Line 160: "solution" should be "solutions" Done (line 187).

[Figure]

- Line 162: remove comma after "H2O2" Done (line 188).

- Line 166-169: should be re-phrased, also I imagine you do not mean that you are looking for "biomarkers" but instead "tracer compounds"? Done (line 194).

- Line 171: define "ESI" the first time you use it Done (see at line 124).

- Line 176: Orbitrap should be capitalized consistently throughout the text Done.

- Line 185: the plural of "spectrum" is "spectra" Done (line 210).

- Line 208: please write out the full name of the "nitrogen rule" Done (line 234).

Results and Discussion - Line 215: title seems incomplete, missing the word "matter" The title is replaced with: 3.1 Mass spectral characteristics of WSOC extracts from WSBA samples (line 241).

- Line 216: "2.5" should be written as a subscript here and throughout the text (i.e.PM2.5) Done (see at line 141 and 242).

- Line 224: "spectrum" should be "spectra" if the authors meant it to be plural, also what is meant by "abstracted blank"? I assume this should say "subtracted" Done (line 254).

- Line 234-235: this sentence is incomplete We use "in spite of" instead of "although", and remove the comma (see line 263).

- Line 265: this sentence should be clarified: what is meant by "low content"? Low mass loading? Low mass loading (line 294).

- Line 290: remove "emerged" Done (line 321).

- Line 320: remove comma after "4h". Done (line 353).

- Line 328: add "after" before the word "photooxidation" Done.

- Line 321-331: should be clarified, if phenols and methoxyphenols are undergoing photochemical aging, how are they impacting POA? The paragraph is re-arranged (line

376-386), and the sentence is deleted.

- Line 351: remove comma after "mechanism". Done (line 381).

- Line 357: for clarity, consider using "chromatograms" instead of "diagrams", and propagate through text. Done (line 404).

- Line 378: remove "emerged". Done (line 423).

- Line 380: remove "stemmed". Done (line 424).

- Line 505: "parameter" should be plural. Done (line 548).

- Line 520: should be re-phrased for clarity, what is meant by "increased tendency", what exactly is "consistent with LC observations"? We have re-phrased it, see line 562-565.

Please also note the supplement to this comment:
https://www.atmos-chem-phys-discuss.net/acp-2019-608/acp-2019-608-AC1-supplement.pdf

---

## Author Comment (AC2) · 2 Feb 2020

All authors would like to thank the reviewer for the constructive comments. These comments will surely improve our manuscript. Below are the point by point the answers to the reviewer comments.

This is an interesting manuscript that describes photo-oxidation of both representative "real" biomass burning organic aerosol (straw) as well as a simpler surrogate containing. I find the paper hard to follow. This is in part because the written English, while

[Figure]

passable, is imperfect. However, the main issue is that there is no clear story beyond "oxidation of WSOC increases the oxidation state" (which is almost guaranteed) and "even a model system from a single fuel is very complex". We have realized the deficiency in writing and further improved the structure of this paper. The results and discussion include: 3.1 Mass spectral characteristics of WSOC extracts from WSBA samples. 3.2 Mass spectral characteristics of the products from photooxidation of phenolic compounds in the aqueous phase. 3.3 Comparison of the photochemical products of phenolic compounds and the CHO composition in WSOC extracts from WSBA samples. 3.4 Photolysis of WSOC extracts from WSBA samples

As far as I can tell a manuscript with a more clearly articulated story would be appropriate for ACP, but I am also on the edge of the subject area, and so a reviewer with greater experience with high-resolution GC methods might be more appropriate. Some general comments: The "/" in "O/C" really means ratio, so "O/C ratio" is redundant. I suggest writing "the oxygen to carbon ratio (O/C)" once and then omitting "ratio" when subsequently using the abbreviation. In this study, we write "the oxygen to carbon ratio (O/C)" once and then use O/C or O/C values to replace it .

There is room in the literature for complex analyses of complex systems and we can not always demand an incredibly simple story, but the paper could still benefit from a major re-write to pull the most important themes to the surface. We have improved the structure of this paper, especially we added Section 3.3: Comparison of the photochemical products of phenolic compounds and the CHO composition in WSOC extracts from WSBA samples.

I do not believe it is appropriate to end the abstract with "accounting for the highly oxygenated nature". Perhaps "contributing to" is warranted but the implication of the current wording is that the contribution dominates, and that has not been demonstrated here. Done. We use "contributing to" to replace "accounting for" at the end of the abstract.

[Figure]

Specific comments: Line 252 "is prone for" is not quite right. "is most sensitive to molecules containing polar ..."? We have rephrased the sentence (see line 279-283).

Line 287 "the all extract samples" ???? Either the authors actually mean "the all-extract samples" or they may mean "all of the extracted samples". Clarify. Here we have deleted the word of "all" (see line 318).

Line 290 "the emerged O/C ratios" could be "the measured O/C values" ("values" is appropriate after O/C in my opinion). Here we use "the measured O/C" to replace "the emerged O/C ratios" (see line 321).

Line 336 "as well as to increased" clashes with the subject "would result in" before, so "to" should be "in". Done (see line 365).

Line 347 "pathway for the low-volatility" strike "the". Deleted (see line 377).

Line 532 "experience the similar" again strike "the". Deleted (see line 582).

Line 543 "aerosols have the potential" the subject is "fraction" so should be "has". Done (see line 596).

Line 544 "partly account for" is better than the abstract but "contribute to" would be best in both places. Here and the abstract we use "contribute to" (see line 52 and 597).

Please also note the supplement to this comment:
https://www.atmos-chem-phys-discuss.net/acp-2019-608/acp-2019-608-AC2-supplement.pdf

———————————————

---

## Author Response (AR2)

**Reply for Reviewer's Comments (Report #1)**

**Thanks for the positive comments on our submission. We would like to further improve our manuscript following the concrete suggestions.**

*Line 276: Electrospray ionization does not ionize CHS compounds particularly well, which the authors state. However, CHN compounds are ionized very well in positive mode (while this study only uses negative mode). This should be clarified here. The way it reads now, it sounds like CHN compounds are never ionized well with electrospray, which is not correct.*

**Line 276-278: We have rephrased the sentence, as follows:**
**"In general, CHN and CHS compounds are not ionized well in negative ESI mode, which could be a reason why these species were not the most prevalent compounds in this study."**

*Line 313: In the discussion of DBEs for CHO and CHON compounds, the authors should note that the double bonds could be present in the O- and/or N- containing functional group OR in the carbon structure. So DBE for CHO or CHON compounds does not necessarily mean that there is a double bond or ring in the carbon structure (it could, but it could also mean*

*that there is a double bond in the functional group), and thus does not necessarily mean that these compounds show aromaticity. With higher DBE values, actual aromaticity becomes more likely, but at lower DBE values, DBE cannot necessarily be equated to carbon structure double bonds or rings.*

**Thanks for the constructive comment. We have rephrased the sentence (line 312-317) as follows: "The average double bond equivalent (DBE) showed relatively high values, 5.5 for CHO compounds and 6.1 for CHON compounds (Table S2), suggesting that unsaturated organic compounds were abundant in the present samples, and their presence could partially account for the strong light-absorbing feature in the near-UV region as observed in our previous study (Cai et al., 2018)."**

*Line 318: Consider reporting average +/- standard deviation as well as the range?*

**Done. We have rephrased the sentence (line 318-321) as follows: "Throughout the extract samples, the average H/C and O/C values were ranging from 1.26±0.38 to 1.31±0.40 and from 0.34±0.24 to 0.42±0.29 for CHO compounds, and from 1.19±0.32 to 1.23±0.35 and**

**from 0.28±0.17 to 0.29±0.15 for CHON compounds (Table S2), respectively."**

*Figure 3: It would be helpful to add a few known biomass burning tracers to the figure as anchor points for readers.*

**Done. We added 4 species to Figure 3 as anchor points, see line 335-339 in the revised text.**

*Line 418: Are you able to quantify how much water was emitted? Or mention whether there are any literature estimates of this?*

**Here we do not quantify the content of emitted water. However, previous study have reported that the humidity content in wheat straw was about 13.5% (*Bi et al., 2009*). Undoubtedly, this substantial water contributed to the formation of burning smog.**

**line 422-425: We rephrased as follows: "Considering that a substantial amount of water in plant body (Bi et al., 2009) was discharged during the process of straw combustion, the occurrence of phenolic dimers might indicate that the aqueous phase reactions played an important role in the formation and evolution of emitted aerosol organic composition."**

*Reference (line 638-639):*

*Bi, Y., Gao, C., Wang, Y., and Li, B.: Estimation of straw resources in China, Transactions of the Chinese Society of Agricultural Engineering,*

**Reply for Reviewer's Comments (Report #2)**

**We would like to thank the reviewer for the constructive comments.**

*Line 111, change order of techniques, it should be LC coupled with HRMS*

**Done, see line 111.**

*Line 276-278. Please clarify the logic of this sentence. The reasoning that something is no easily ionized does imply that there is low abundance of it. On contrary, such species may be underestimated.*

**Line 276-278: We have rephrased the sentence, as follows:**
**"In general, CHN and CHS compounds are not ionized well in negative ESI mode, which could be a reason why these species were not the most prevalent compounds in this study."**

*Line 279. The word "prefers" is not suitable for this sentence perhaps use "selectively targets"*

**Done, see line 281.**

*Line 281, Authors state "there were a number of compounds." Were these compounds reported in his paper do authors mean "there are" i.e., stating*

*general fact.*

**We have revised it.**

*Line 282, there should be coma after e.g.*

**Done.**

*Line 569 insert the space between number and unit.*

**Done, see line 574 in the revised text.**

[revised manuscript text omitted]